# A Comprehensive Forest Biomass Dataset for the USA Allows Customized Validation of Remotely Sensed Biomass Estimates

**James Menlove and Sean P. Healey \***

US Forest Service Rocky Mountain Research Station, Ogden, UT 84401, USA; james.menlove@usda.gov

\* Correspondence: sean.healey@usda.gov

**Abstract:** There are several new and imminent space-based sensors intended to support mapping of forest structure and biomass. These instruments, along with advancing cloud-based mapping platforms, will soon contribute to a proliferation of biomass maps. One means of differentiating the quality of different maps and estimation strategies will be comparison of results against independent field-based estimates at various scales. The Forest Inventory and Analysis Program of the US Forest Service (FIA) maintains a designed sample of uniformly measured field plots across the conterminous United States. This paper reports production of a map of statistical estimates of mean biomass, created at approximately the finest scale (64,000-hectare hexagons) allowed by FIA's sample density. This map may be useful for assessing the accuracy of future remotely sensed biomass estimates. Equally important, fine-scale mapping of FIA estimates highlights several ways in which field- and remote sensing-based methods must be aligned to ensure comparability. For example, the biomass in standing dead trees, which may or may not be included in biomass estimates, represents a source of potential discrepancy that FIA shows to be particularly important in the Western US. Likewise, alternative allometric equations (which link measurable tree dimensions such as diameter to difficult-to-measure variables like biomass) strongly impact biomass estimates in ways that can vary over short distances. Potential mismatch in the conditions counted as forests also varies greatly over space. Field-to-map comparisons will ideally minimize these sources of uncertainty by adopting common allometry, carbon pools, and forest definitions. Our national hexagon-level benchmark estimates, provided in Supplementary Files, therefore addresses multiple pools and allometric approaches independently, while providing explicit forest area and uncertainty information. This range of information is intended to allow scientists to minimize potential discrepancies in support of unambiguous validation.

**Keywords:** biomass; forest inventory; validation

## 1. Introduction

Forests play an important role in the global carbon cycle, as the storage of atmospheric carbon in the form of forest biomass significantly influences the planet's radiative balance [1,2]. In the United States, growth of forests mitigates on the order of 15% of national fossil fuel emissions [3]. Carbon storage across forests is far from uniform, however. Biomass gradients occur across the world's forests due to land use conversion [4], varying growing conditions and the impact of forest disturbance [5]. These gradients must be understood if carbon-related ecosystem services are to be measured and managed.

National forest inventories are the international standard for measuring carbon at the country scale [3,6]; their field-based measurements and statistical sample design contribute to straightforward

estimates of biomass and uncertainty. However, the capacity of these inventories to resolve biomass gradients and spatial patterns is limited by the density of their (often expensive) field measurements. Also, inventories are not well developed in many countries, and significant discontinuities occur at many borders because of incompatible definitions and methods.

Space-based remote sensing offers globally consistent forest measurements that address some of these limitations. At least three lidar missions and two radar missions will soon be providing high-quality forest structure information relevant to biomass measurement. Lidar instruments include the already launched ICESat-2 canopy profiling lidar [7] and the GEDI full-waveform system [8] from NASA as well as the upcoming Multi-footprint Observation Lidar and Imager (MOLI) from JAXA [9]. Planned radar missions include the P-band ESA instrument called BIOMASS [10] and the NASA/ISRO L- and S-band NISAR satellite [11].

In light of the emergence of cloud-based platforms such as Google Earth Engine [12,13], which allow highly parallelized computing and provide a robust application programming interface, there will soon be a variety of forest biomass maps at multiple scales. Judging among alternative maps and choosing the map most appropriate for a particular application will be a complex process. The Land Product Validation (LPV) sub-working group of the Committee on Earth Observation Satellites (CEOS) addresses these factors and highlights the need for open access to reference datasets that follow consistent and straightforward protocols [14]. Duncanson et al. [15] and Bell et al. [16] outline processes for using high-resolution lidar datasets to compare biomass maps from different platforms. However, such datasets are rare and expensive to collect, and they may not be available for many applications.

National forest inventories are collected consistently across large areas and may augment validation options for remotely sensed biomass maps. The inventory of the United States is managed through the Forest Service's Forest Inventory and Analysis Program (FIA) [17]. FIA plot data are collected across the country at field plots selected randomly at a nominal density of one plot per 2428 ha. One way to use this inventory data to evaluate alternative biomass maps is to use the plot data to calibrate independent satellite-based biomass maps [18,19], which may be compared against candidate biomass maps. However, while definitions in these maps are consistent with FIA protocols, they are subject to many of the same sources of error as the maps one would want to validate.

This Technical Note documents an alternative use of inventory data for validation. We produced a map of local-scale statistical estimates, based solely on a designed sample of ground measurements, that may be of use in evaluating the myriad remotely sensed biomass maps that will soon be in circulation. Brown and Schroeder [20] used FIA's sampling frame to make statistical estimates of biomass in the Eastern US at the level of the county. These estimates provide the opportunity to evaluate biomass levels implied by remote sensing-based methods against design-based, field-calibrated estimates across hundreds of diverse counties. The current paper extends this idea to the entire conterminous United States, and provides biomass estimates at what is generally considered to be the finest spatial grain supported by FIA's standard sample intensity. Specifically, we map FIA estimates of biomass using a hexagonal grid that includes at least 25–30 plots per grid cell (or "hex"). The primary purpose of this technical note is to document this spatial dataset, which provides 9876 local non-zero (12591 total) estimates of mean biomass against which alternative remotely sensed estimates might be evaluated.

These "benchmark" estimates are considered more authoritative than the remotely sensed estimates to which they may be compared because they are based solely upon straightforward sample theory and quality-controlled field measurements instead of models using auxiliary data. This is not to say, however, that the inventory-based benchmark is without uncertainty. The second goal of this paper, beyond documenting an FIA-based reference dataset, is to illuminate concurrent sources of uncertainty that complicate the task of assessing the error of a remotely sensed biomass map. For example, biomass is rarely directly measured in the field; FIA and other national forest inventories use allometric models to convert standard non-destructive measurements such as crown height and bole diameter into "measurements" of biomass. Zhao et al. [21] found that calibrating lidar-based models with identical ground data subject to different allometric models can yield maps that vary

substantially, particularly in high-biomass areas. Vorster et al. [22] found that uncertainty due to allometric error can exceed remote sensing error in the production of biomass maps with Landsat data. Using FIA's sampling frame and applying different sets of equations to plot measurements of height and diameter, we map differences in statistical estimates of biomass due solely to the choice of allometry.

Another source of uncertainty affecting benchmark validation using FIA data is the choice of carbon pools considered. Many maps identify aboveground biomass density (AGBD, in mass per hectare units) as the variable of interest. Calibration data may or may not include standing dead trees. Since this pool can represent a significant but highly variable source of biomass [23], its inclusion or exclusion from the reference data used to calibrate a biomass map may play an important role in how well the map matches benchmark biomass estimates. We use FIA estimates to quantify the impact of standing dead trees on total aboveground carbon stocks across the US.

Disagreement about the forestland base over which biomass should be estimated is another source of uncertainty in the validation process. Inventory-based estimates of biomass density tend to restrict estimates to forestland (e.g., [24]), which is a strictly defined class using variables such as stocking potential that are difficult to measure from space. However, trees outside of these areas—in urban or rangeland areas, for instance—can represent substantial amounts of biomass [25]. Forest cover maps can be used to mask out non-forest trees, but cover maps can differ both from each other and from inventory-based forest area estimates. Consequently, the issue of which trees are considered in estimates of mean and total biomass can become an important source of uncertainty [26]. Our hex-level analysis of FIA data identifies areas where mixing of forest and non-forest conditions may make forest mask issues more important.

Lastly, field sampling error complicates validation of biomass maps against inventory-based estimates. FIA estimates are presumed to be unbiased, but their uncertainty varies widely across the country as a function of both sample intensity (which can vary by state) and natural variation in the underlying forest population. We depict the uncertainty of FIA biomass estimates at the hex level to allow a more nuanced comparison of maps and inventory data. Agreement between maps and the FIA benchmark within hexes having lower FIA uncertainty may be more important from a validation perspective than agreement in areas where FIA's confidence interval is broader. The intent of this investigation is to not only document FIA estimates that may be used as a benchmark for alternative remotely sensed maps, but also to draw from the inventory factors that expand consideration of uncertainty and provide context for the use of inventories to validate remotely sensed biomass estimates.

## 2. Materials and Methods

### 2.1. Environmental Monitoring and Assessment Program (EMAP) Hexagons

White et al. [27] described a hierarchical hexagonal tessellation of the Earth's surface based upon decomposition of a triangular network of sample points. The distance between these points was approximately 27 km, implying approximately equal-area hexagons of 640 km$^2$. Positioning of this tessellation, referenced here as the EMAP hexagons, was optimized for coverage of the United States. FIA uses a similar but spatially offset and intensified tessellation to allocate plots [17], such that there are approximately 27 FIA plots within each EMAP hex. It should be noted that some stakeholders (e.g., individual states) have augmented plot collection using finer-scale hexagons, which supports inventory analysis at finer scales. Estimation at the level of the EMAP hex has provided a balance between fine-scale spatial detail and inclusion of enough plots (around 27) to support relatively precise estimation (e.g., [28]).

### 2.2. FIA Measures of Biomass

FIA's national network contains nearly 327,000 plots, approximately one third of which fall in forested areas. Each plot is randomly assigned to a "panel" representing a fixed fraction of

the sample (10–20%, depending on the state), and one panel is typically measured every year. Standard design-based statistical estimators are used to make estimates of forest conditions at different scales. These estimators are described comprehensively elsewhere [17,29,30] and are not detailed in this Technical Note. Because plots are measured using a rolling panel design, estimates represent conditions associated with a lagging time window of 5–10 years, depending on the sample fraction assigned to each panel.

Several standard FIA forest biomass variables [29], described below, were estimated for each EMAP hexagon. Estimates for these variables, summarized in Table 1, were stored by hex in a geospatial database that is included here as Supplementary Materials. Estimates were expressed as a ratio of biomass per unit of sampled land. The denominator of this ratio was the FIA estimate area of total land, excluding water and foreign territory (EST_SAMPLED_HA in Table 1). For most hexagons, the estimated biomass ratio equaled the total biomass estimated for forests within the hexagon divided by the entire area of the hexagon. Biomass was estimated in this way to support the broadest possible use of the hex-level estimates to evaluate the accuracy of biomass maps and to minimize the impact of forest definition discrepancies.

**Table 1.** Attributes estimated for each EMAP hex using the FIA database.

| | |
|---|---|
| EMAP_HEX | FIADB EMAP hexagon identifier. The same as USHEXES_ID in the original EMAP GIS layer. |
| PROP_FOREST | Estimate of proportion of the area that is forest land. Ratio estimate of forest land area over sampled area. Unitless. |
| SE_PROP_FOREST_PCT | Sampling error of estimate of the forest land proportion, as a percent of the estimate. |
| CRM_LIVE | Estimate of aboveground biomass of live trees (≥2.54 cm diameter) on forest land per hectare of sampled area, using FIA component ratio method (CRM). Megagrams per hectare. |
| SE_CRM_LIVE_PCT | Sampling error of CRM live biomass per hectare estimate, as a percent of the estimate. |
| CRM_STND_DEAD | Estimate of aboveground biomass of standing dead trees (≥12.7 cm diameter) on forest land per hectare of sampled area, using FIA component ratio method (CRM). Megagrams per hectare. |
| SE_CRM_STND_DEAD_PCT | Sampling error of CRM standing dead biomass per hectare estimate, as a percent of the estimate. |
| CRM_LIVE_DEAD | Estimate of aboveground biomass of live trees (≥2.54 cm diameter) plus standing dead trees (≥5 inches diameter) on forest land per hectare of sampled area, using FIA component ratio method (CRM). Megagrams per hectare. |
| SE_CRM_LIVE_DEAD_PCT | Sampling error of CRM live plus standing dead biomass per hectare estimate, as a percent of the estimate. |
| DRYBIOT_LIVE | Estimate of aboveground biomass of live trees (≥2.54 cm diameter) on forest land per hectare of sampled area, using retired FIA regional methods. Megagrams per hectare. |
| SE_DRYBIOT_LIVE_PCT | Sampling error of regional method live biomass per hectare estimate, as a percent of the estimate. |
| DRYBIOT_STND_DEAD | Estimate of aboveground biomass of standing dead trees (≥12.7 cm diameter) on forest land per hectare of sampled area, using retired FIA regional methods. Megagrams per hectare. |
| SE_DRYBIOT_STND_DEAD_PCT | Sampling error of regional method standing dead biomass per hectare estimate, as a percent of the estimate. |
| DRYBIOT_LIVE_DEAD | Estimate of aboveground biomass of live trees (≥2.54 cm diameter) plus standing dead trees (≥5 inches diameter) on forest land per hectare of sampled area, using retired FIA regional methods. Megagrams per hectare. |

**Table 1.** *Cont.*

| | |
|---|---|
| SE_DRYBIOT_LIVE_DEAD_PCT | Sampling error of regional method live plus standing dead biomass per hectare estimate, as a percent of the estimate. |
| JENK_LIVE | Estimate of aboveground biomass of live trees (≥2.54 cm diameter) on forest land per hectare of sampled area, using Jenkins equation. Megagrams per hectare. |
| SE_JENK_LIVE_PCT | Sampling error of Jenkins live biomass per hectare estimate, as a percent of the estimate. |
| JENK_STND_DEAD | Estimate of aboveground biomass of standing dead trees (≥12.7 cm diameter) on forest land per hectare of sampled area, using Jenkins equation. Megagrams per hectare. |
| SE_JENK_STND_DEAD_PCT | Sampling error of Jenkins standing dead biomass per hectare estimate, as a percent of the estimate. |
| JENK_LIVE_DEAD | Estimate of aboveground biomass of live trees (≥2.54 cm diameter) plus standing dead trees (≥12.7 cm diameter) on forest land per hectare of sampled area, using Jenkins equation. Megagrams per hectare. |
| SE_JENK_LIVE_DEAD_PCT | Sampling error of Jenkins live plus standing dead biomass per hectare estimate, as a percent of the estimate. |
| EST_SAMPLED_HA | Estimate of sampled hectares in the hexagon. |
| SAMPLED_PLOTS | Number of sampled plots in the hexagon. |
| NON_SAMPLED_PLOTS | Number of non-sampled plots in the hexagon. |

Specifically, FIA's field-based definition of forestland considers variables that are difficult to observe remotely [31] and that are not available in a wall-to-wall map. If the user's biomass map does not use an explicit forest/non-forest mask, their average pixel-level prediction of biomass may be compared directly with the FIA biomass estimates described here. If the user is interested in the biomass stored only in the areas of mapped forest within the hexagon, the FIA number described here may be divided by the fraction of forest cover implied by the user's forest mask. For context, this fraction may be compared to the FIA estimate of proportion of forest within the hex, which was estimated separately and is available in conjunction with the biomass ratio described above.

FIA's nationally consistent plot design is composed of four 0.017-hectare subplots, on which measurements of every tree (≥12.7 cm diameter) are recorded. Measurements include variables such as height, species, and diameter at breast height and are subject to well-defined measurement quality objectives (Pollard et al., 2005). Direct observation of biomass through destructive sampling is not feasible across large inventories, so allometric biomass models are used for every tree to scale observable tree dimensions such as diameter and height to biomass. Three different sets of allometric equations are available in FIA's database, and we derived separate hex-level biomass estimates from each set.

Allometry under these three variations is specific by species or species group. The first set of models, based on tree diameter measurements alone, was published by Jenkins et al. [32] and is here called the "Jenkins" allometry. A second approach, called the Component Ratio Method (CRM: [33]), is based upon FIA estimates of sound bole volume, with other tree components estimated as ratios [32] of computed bole biomass. CRM is currently used by FIA for national carbon reporting. Unlike the Jenkins equations, CRM makes use of height measurements (through computation of volume). Estimates were also made using an FIA variable called DRYBIOT, a retired FIA biomass variable that was evaluated to support potential calibration against legacy FIA analyses. It should be noted that while FIA's measurements of CRM dead-tree biomass include a biomass reduction factor linked to decay class [23], to the best of our knowledge there is no such factor applied for biomass in dead trees calculated with the Jenkins and DRYBIOT variables. Applying different allometries to tree-level measurements creates varying plot-level measurements, and that variance propagates to

statistical estimates of biomass at larger scales. Here, we map differences in hex-level biomass estimates that are traceable to the difference between use of CRM and Jenkins allometries.

Three pool-specific estimates were made for each hex under each allometry, covering biomass in: (1) live trees; (2) dead trees; and (3) the combination of live and dead trees. Sampling error was assessed for each mean biomass estimate using standard FIA design-based estimators. FIA's plot network is treated as a simple random sample, and exhaustive description of the estimators used by the Inventory are available elsewhere [17,29,34]. Hex-level estimates of each of the variables in Table 1 were written to a spatial database covering the conterminous United States (included in Supplementary Materials).

## 3. Results

The spatial distributions of hex-level estimates of several key variables are summarized below and documented in a geodatabase file and a similar kml file (with truncated variable names), both of which is stored in the Supplementary Data. Recall that the biomass density estimates shown in Figure 1 and Figure 4, as well as the standard error estimates in Figure 2 and the difference between CRM and Jenkins allometries in Figure 5, all consider biomass as a ratio of measured forest biomass divided the entire sampled area of the polygon.

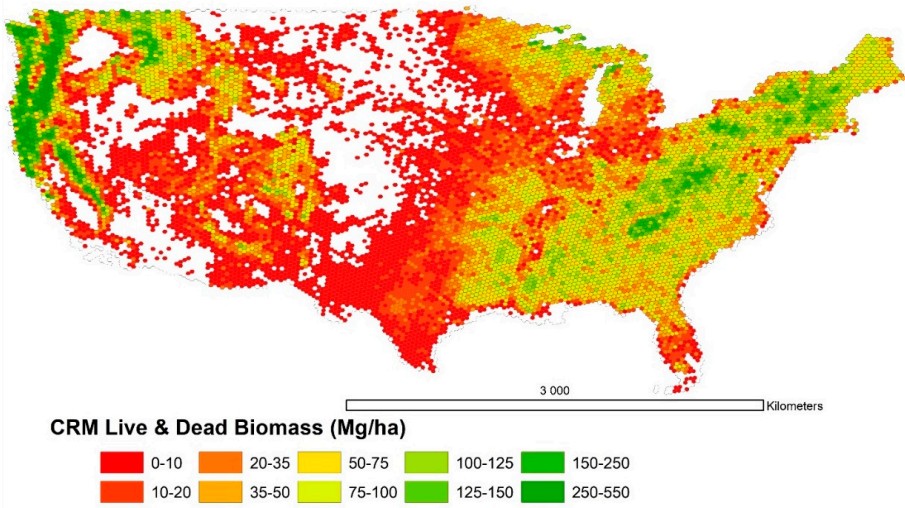

**Figure 1.** FIA hex-level estimates of CRM (Component Ratio Method) biomass.

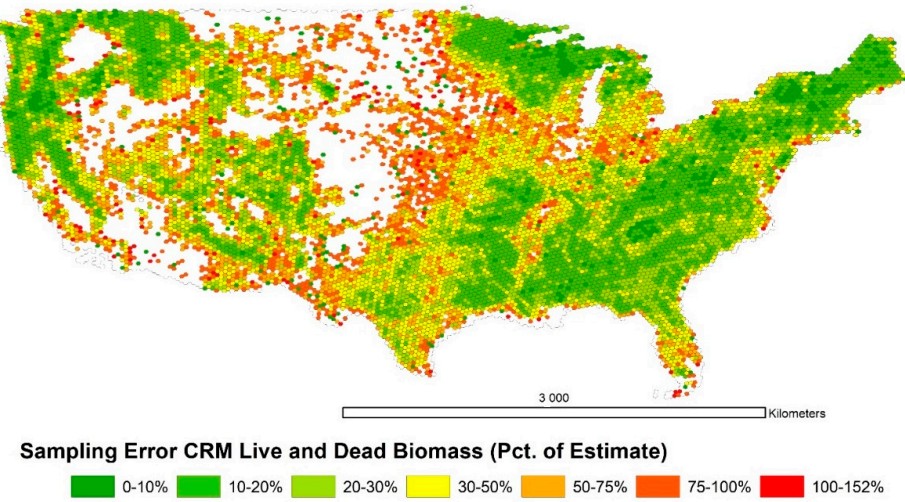

**Figure 2.** FIA hex-level estimates of the standard error of mean CRM biomass.

The highest levels of biomass occur in the forests of the Northwest, while the lowest values occur in the country's interior, in areas where forested conditions make up only a small proportion of the polygon. Areas of highest uncertainty, as a percentage of the estimate (Figure 2), occur in areas where the proportion of forest is low (Figure 3). This is both because variation of biomass in these areas is high and because mean values are low, which inflates the standard error when it is expressed as a percentage of the estimate.

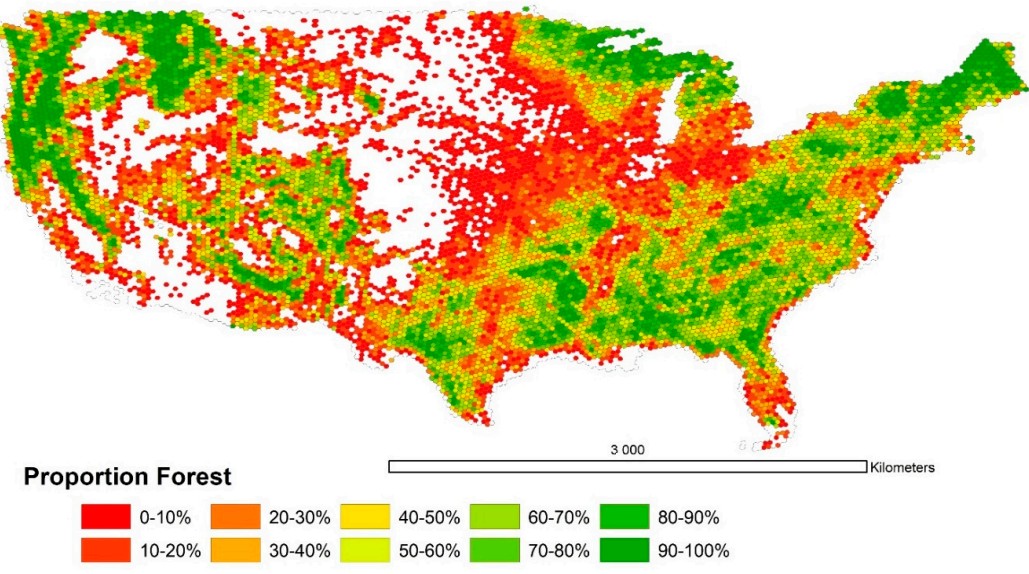

**Figure 3.** FIA hex-level estimates of the proportion of forest.

The highest levels of biomass in standing dead trees occur in the western half of the country (Figure 4). In most of the East, standing dead trees were estimated to amount to less than 4 Mg/ha. The effect of allometry was less than 10 Mg/ha in much of the country, but Jenkins equations almost always resulted in higher estimates of biomass density (Figure 5). In extreme cases, this difference amounted to over 80 Mg/ha. CRM-based estimates of biomass were higher only in scattered hex polygons in the Southeast.

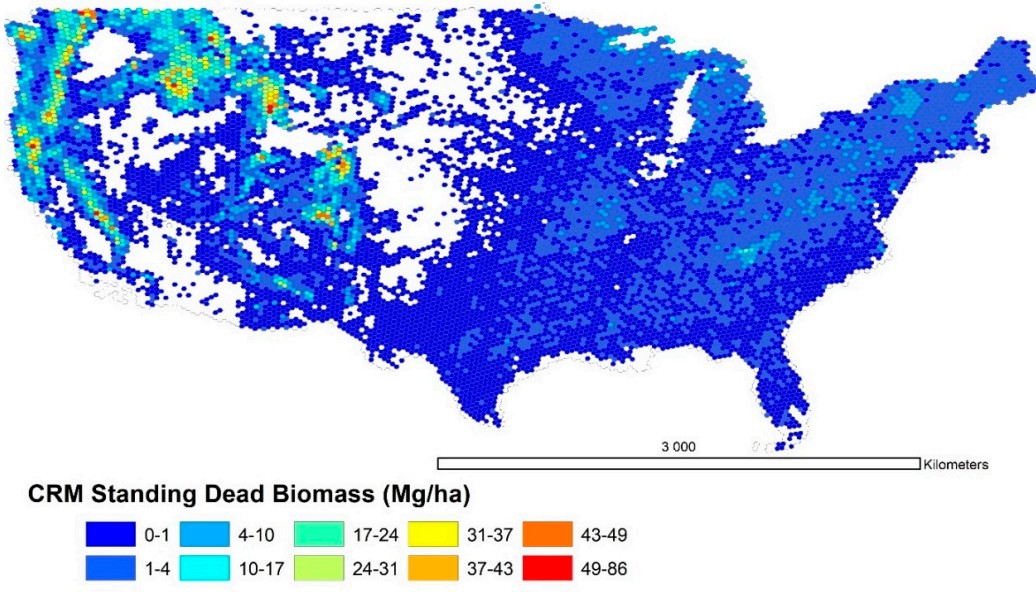

**Figure 4.** FIA hex-level estimates of CRM biomass of standing dead trees.

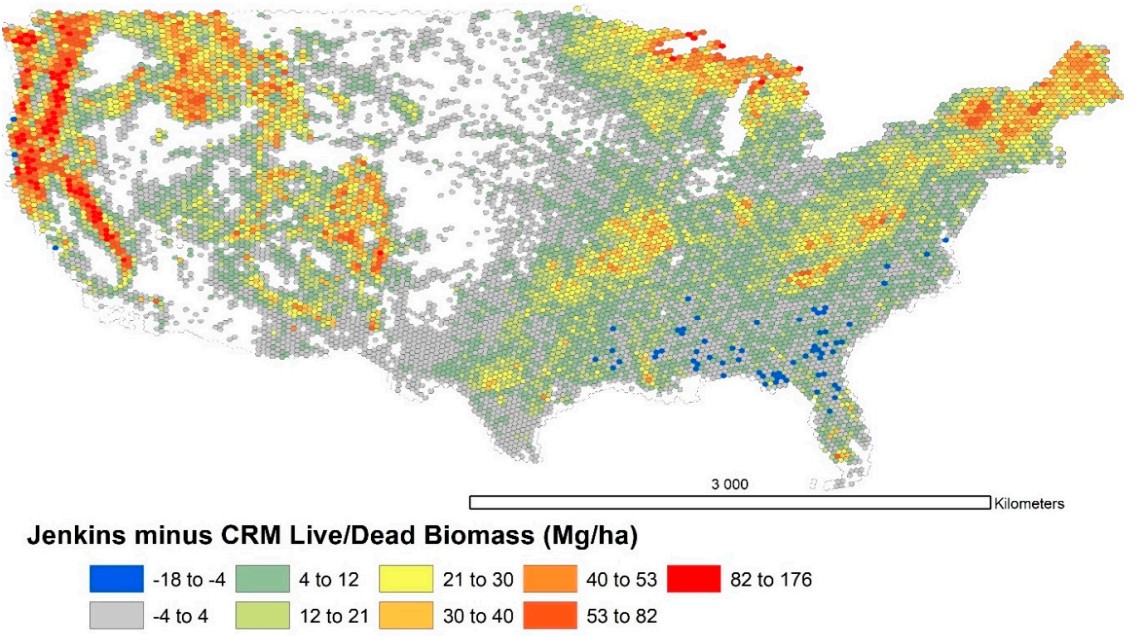

**Jenkins minus CRM Live/Dead Biomass (Mg/ha)**

| | | | | |
|---|---|---|---|---|
| -18 to -4 | 4 to 12 | 21 to 30 | 40 to 53 | 82 to 176 |
| -4 to 4 | 12 to 21 | 30 to 40 | 53 to 82 | |

**Figure 5.** The difference between FIA estimates of live-tree biomass using CRM and the Jenkins equations. Hexagons with greater CRM estimates of biomass are represented in blue.

The average measurement year of FIA plots used in this analysis varied strongly at the regional level (Figure 6); plots in Eastern states are typically measured once every 5–7 years, whereas Western plots are measured on a 10-year cycle. There is also state-level variation in average measurement year. The sample has been augmented in some states by FIA stakeholders, resulting in intensified data collection and hex-level estimates more representative of present condition. In general, one would expect the average plot-level inventory year to advance one year annually, given FIA's annual sample design. Some disruptions in plot measurement are expected, however, for the current year due to the Covid-19 pandemic.

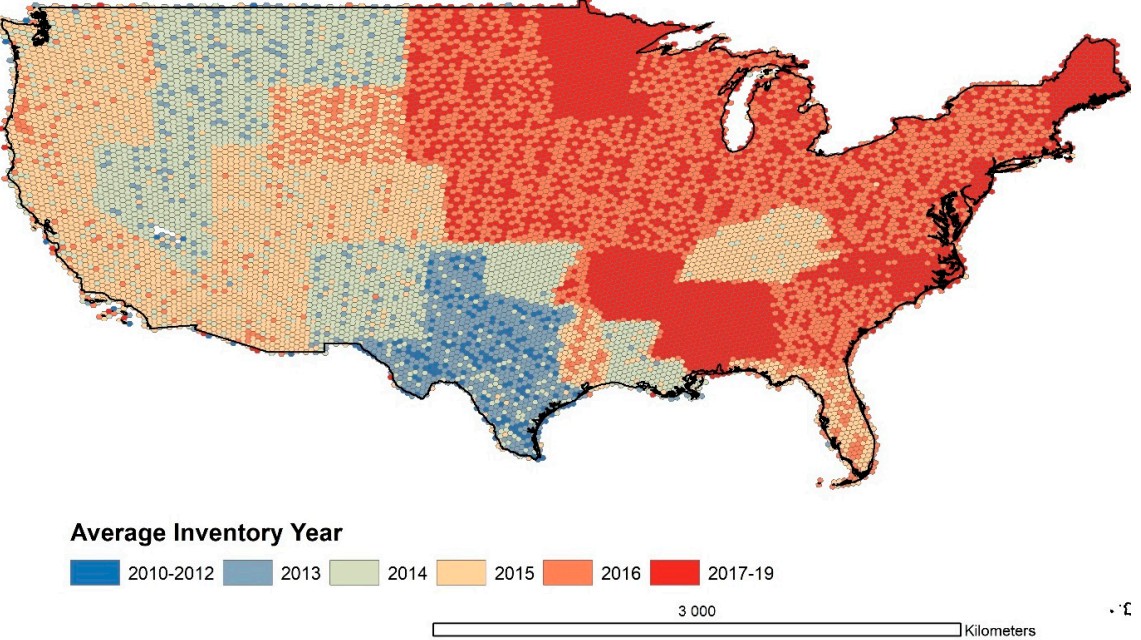

**Average Inventory Year**

| | | | | | |
|---|---|---|---|---|---|
| 2010-2012 | 2013 | 2014 | 2015 | 2016 | 2017-19 |

**Figure 6.** The average year of data collection for the inventory data used in the FIA estimates presented in this paper.

## 4. Discussion

This analysis provided fine-scale biomass benchmark information over the United States (Figure 1), against which remote sensing-based means of estimating biomass might be evaluated. Remotely sensed estimates are increasingly derived through formal modes of inference and are accompanied by their own estimates of variance. These methods may make use of properties of the model-building process and a wall-to-wall map (e.g., [35]), or they may be based upon only a sample of high-quality remote sensing data, integrating both modeling and sampling uncertainty (e.g., [36]). Statistical estimators also exist for scaling up across a hierarchy of remotely sensed data [37]. FIA estimates have been widely used to provide context around biomass levels implied by remote sensing at county, state, and national scales [38–41].

While the current analysis standardizes the process of evaluating spatial biomass products at the finest possible scale, it also highlights the need for circumspection when comparing inventory and remotely sensed data. Figure 2 indicates that the certainty of FIA's estimates at the hex level varies substantially; in some hex units, large FIA sampling error reduces what we can learn about the accuracy of biomass levels predicted through remote sensing. However, it should be kept in mind that FIA's sample is presumed to be unbiased [17]; when assessed over a large number of hexes, one would not expect a similarly unbiased map to produce hex-level estimates either consistently above or below the FIA estimates.

The allometric equations used to assign a biomass to trees measured in the field were also shown to be an important consideration. Domke et al. [42] found that the estimate of national carbon stocks decreased by 16% when FIA moved from a set of generalized equations from Jenkins et al. [32] to the currently used component ratio method (CRM). Duncanson et al. [43] found similar discrepancies at the county scale. When using FIA estimates as a benchmark, it is important to remember that many remotely sensed biomass maps are often global in scope and are therefore unlikely to be calibrated consistently with the state-specific volume equations that comprise the core of CRM. Comparison of Figures 1 and 4 shows that, even when using an identical sample of trees, allometry can cause population estimates to vary 50% or more at the level of the hex. Unlike sampling error, allometric discrepancies are likely to cause systematic differences between maps and inventory estimates.

The issue of biomass in standing dead trees can also complicate comparison of map- and inventory-derived biomass estimates. Large stocks of standing dead trees in the Rocky Mountains (Figure 4) are partially due to recent outbreaks of mountain pine beetle (*Dendroctonus ponderosae*), which in many areas have killed a majority of large trees [44]. Dry forests throughout the West have been the site of increasing wildfire [45]. Drier landscapes in the West also have lower autotrophic respiration rates [46], which likely increases the duration of disturbance-killed trees prior to decomposition.

FIA explicitly measures the biomass in such trees (Figure 4), but remote sensing methods often do not. Standing dead trees can be predicted as a function of stand structure using active sensing technologies like lidar [47], but it is difficult to directly distinguish live from dead trees using height return data [48]. Maps created with optical sensors can be sensitive to canopy mortality [49], but to date we are unaware of research with sensors such as Landsat that differentiates dead- from live-tree biomass. Spatially explicit inventory estimates, such as those presented here, may help map producers decompose overall map error by highlighting uncertainty related to dead trees.

Differences in the recency of field and remote sensing data represent a seemingly straightforward caveat in the error assessment process. Disturbance can generate large, immediate changes in live biomass [50], and attention to timing is needed when inventory data are matched with maps. Figure 6 shows how complicated this process can be, however. Operational differences in the inventory can result in different-vintage field estimates even within the same state. There is likewise complexity in simply determining the conditions over which biomass estimates are made and compared. Varying definition of forests, and the potential detection of trees outside of forests, can produce divergence in map- and inventory-based estimates that has nothing to do with the quality of remotely sensed predictions [41]. As mentioned above, the FIA dataset documented here supports

efforts to understand the role of forest/non-forest confusion by including hex-level FIA estimates of the proportion of forest.

## 5. Conclusions

New spaceborne instruments and the evolution of cloud-enabled mapping platforms will likely precipitate a proliferation of broad-scale forest biomass maps and estimation tools. Large, ground-based inventories provide an opportunity to validate maps in a consistent way across many ecological settings. We used FIA (the national forest inventory of the United States) to produce consistentestimates of biomass and other relevant variables across 12,591 local hexagonal areas (Supplementary Files). Most directly, these field-based estimates represent a benchmark against which remote sensing scientists may evaluate the accuracy and potential biases of space-based predictions or estimates of biomass. More broadly, this dataset highlights potential discrepancies that must be addressed to reduce ambiguity in the validation process.

One such discrepancy involves timing. Measurement latency varies spatially in FIA's sample, and in places where the time between field measurements and remote sensing measurements is high, users should be alert to artifacts caused by any intervening large disturbances. Discrepancies may also result from factors such as the inclusion/exclusion of dead trees, the presence of which varies strongly by region, and from differences in how forestland is defined. We also showed that divergent choices about allometry can introduce substantial disagreement even when using exactly the same measurements and statistical estimators. Our inventory-based maps suggest that the effect of these factors varies in complex ways over short distances. In constructing an FIA-based validation dataset, we therefore included a range of variables that should allow the user flexibility in matching the benchmark to the remotely sensed product to be validated. Specifically, the dataset attached as Supplemental Data includes estimates of different combinations of live and dead tree biomass across three different sets of allometry. The dataset is also explicit at the level of the local hexagon about the estimated proportion of forest and the uncertainty of all variants. It is hoped that this dataset may be useful both for immediate assessment of biomass predictions and for broadening the discussion around the use of field data to validate remote sensing.

**Supplementary Materials:** The following are available online at https://zenodo.org/record/4294490# .X9o5Q9hKg2w: FIA Benchmark Biomass Estimates for sub-regions of the United States (geodatabase and kml formats).

**Author Contributions:** Conceptualization, S.P.H. and J.M.; methodology, J.M.; software, J.M.; data curation, J.M.; writing—original draft preparation, S.P.H.; writing—review and editing, J.M. and S.P.H. visualization, S.P.H.; supervision, S.P.H.; project administration, S.P.H.; funding acquisition, S.P.H. All authors have read and agreed to the published version of the manuscript.

**Funding:** This work was supported in part by the U.S. Department of Agriculture Forest Service. Funding was also provided by NASA Carbon Monitoring System Grant #80HQTR18T0016 (Healey).

**Conflicts of Interest:** The authors declare no conflict of interest.

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
