# Peer review of "A Comprehensive Forest Biomass Dataset for the USA Allows Customized Validation of Remotely Sensed Biomass Estimates"

_remotesensing, doi:10.3390/rs12244141_

Round 1

Reviewer 1 Report

Introduction

  1. Full stop should be added at Line No 28.
  2. The objectives are not featuring clearly, the motivation should be build along the line of the specific objectives not broader story

Materials

Line 162-163 Sampling error was assessed for each mean biomass estimate using standard FIA design-based estimators, as described
in the sources cited above. There are so many Design based estimators, it could be of interest to the estimators to be presented, showing how you input your data on this. Furthermore, is important to describe how your sampling design AGREES with the use of these estimators as each has its own requirement.

Results

Line 190. The authors are reporting the effect of alometry, however in the methodology is not explicit shown how this effect was computed

Again the results from the statistical estimators say variance using GREG is not featuring

It will be of interest to compute relative efficiency R.E (read sandras book for more info on this and how to articulate in your work)

Author Response

Comment Revision
Full stop should be added at Line No 28. Done
The objectives are not featuring clearly, the motivation should be build along the line of the specific objectives not broader story This statement was added to the Introduction for clarification: "This Technical Note documents an alternative use of inventory data for validation.  We produced a map of local-scale statistical estimates, based solely on a designed sample of ground measurements, that may be of use in evaluating the myriad remotely sensed biomass maps that will soon be in circulation."  There is also expanded explanation of the second objective, which is summarized with this statement: "The second goal of this paper is to illuminate the effects of concurrent sources of uncertainty that complicate the task of assessing the error of a remotely sensed biomass map."
Line 162-163 Sampling error was assessed for each mean biomass estimate using standard FIA design-based estimators, as described in the sources cited above. There are so many Design based estimators, it could be of interest to the estimators to be presented, showing how you input your data on this. Furthermore, is important to describe how your sampling design AGREES with the use of these estimators as each has its own requirement. The sampling design of FIA (the US national forest inventory) is now specified as a simple random sample, and additional references detailing methods have been added.  Also, the entirety of Section 2.2 provides details about field methods and statistical estimators used by the Inventory (and in our inventory-based hex-level estimates).  Substantial information has been added to that section.
Line 190. The authors are reporting the effect of alometry, however in the methodology is not explicit shown how this effect was computed The section about allometry has been clarified, and this statement was added: "Applying different allometries to tree-level measurements creates varying plot-level measurements and propagates to statistical estimates of biomass at larger scales.  Here, we map differences in hex-level biomass estimates that are traceable to the difference between use of CRM and Jenkins allometries."
Again the results from the statistical estimators say variance using GREG is not featuring GREG models use auxiliary data, and I am wondering if the reviewer might think we are using remote sensing to create these maps.  Hopefully after the above clarifications, it will be clearer that we made inventory-based estimates to: 1) support validation of other people's maps, and 2) understand uncertainties involved with comparing inventory and map-based estimates.
It will be of interest to compute relative efficiency R.E (read sandras book for more info on this and how to articulate in your work) Relative efficiency is measure of precision improvement when ancillary data (often from remote sensing) is used to stratify design-based estimates.  Similar to the note above, this was not the purpose of this paper.
  Thank you for the feedback and helpful suggestions

Reviewer 2 Report

The title “Understanding uncertainty of remotely sensed forest biomass estimates with a field-based benchmark product for the United States”  leads to a mistaken impression that the focus of the manuscript is the analysis of uncertainty in the biomass estimation by remote sensing. The paper reports, in fact, the production of a reference base from in situ sample data obtained systematically by the USFS in the FIA Program, which has the potential to calibrate the various biomass maps at the most diverse scales.

Although it is undeniable the relevance of documenting the production of this reference, including the various aspects related to the accuracy of the information produced, I believe that it is necessary to introduce a validation experiment, so that the manuscript becomes more adherent to the scope of the journal and the discussion on the accuracy aspects of the maps become concrete and empirically based. That is, the development of an experiment that results in the biomass estimation by remote sensing for a test area, would allow to validate the database and make the manuscript more adherent to the scope of Remote Sensing (even if it is a Technical Note).

As it stands, the manuscript is not suitable for publication in Remote Sensing. In addition, the methods do not present enough details to understand the approach adopted and the results do not directly support the discussions made.

Further considerations about the submitted manuscript were made in the manuscript attached.

Author Response

Comment Revision
The title “Understanding uncertainty of remotely sensed forest biomass estimates with a field-based benchmark product for the United States”  leads to a mistaken impression that the focus of the manuscript is the analysis of uncertainty in the biomass estimation by remote sensing. The paper reports, in fact, the production of a reference base from in situ sample data obtained systematically by the USFS in the FIA Program, which has the potential to calibrate the various biomass maps at the most diverse scales.  Although it is undeniable the relevance of documenting the production of this reference, including the various aspects related to the accuracy of the information produced, I believe that it is necessary to introduce a validation experiment, so that the manuscript becomes more adherent to the scope of the journal and the discussion on the accuracy aspects of the maps become concrete and empirically based. That is, the development of an experiment that results in the biomass estimation by remote sensing for a test area, would allow to validate the database and make the manuscript more adherent to the scope of Remote Sensing (even if it is a Technical Note).   As it stands, the manuscript is not suitable for publication in Remote Sensing. In addition, the methods do not present enough details to understand the approach adopted and the results do not directly support the discussions made We acknowledge that the goals and message of the paper were not sufficiently clear in the original submission.  Both objectives mentioned by the reviewer (introduction of a reference dataset and analysis of the uncertainty involved with validating remotely sensed product using ground data) are goals of this paper.  Several alterations to the Introduction make this more clear.  For example, we added this sentence: This Technical Note documents an alternative use of inventory data for validation.  We produced a map of local-scale statistical estimates, based solely on a designed sample of ground measurements, that may be of use in evaluating the myriad remotely sensed biomass maps that will soon be in circulation.
We also edited this statement: "These “benchmark” estimates are considered more authoritative than the remotely sensed estimates to which they may be compared because they are based solely upon straightforward sample theory and quality-controlled field measurements instead of models using auxiliary data.  This is not to say, however, that the inventory-based benchmark is without uncertainty. The second goal of this paper, beyond documenting an FIA-based reference dataset, is to illuminate concurrent sources of uncertainty that complicate the task of assessing the error of a remotely sensed biomass map."  Further, we re-wrote much of both the Abstract and the Conclusion sections to better articulate the what we learned about potential inventory-map discrepancies.  Lastly, we changed the title to: “A comprehensive forest biomass dataset for the US allows customized validation of remotely sensed biomass estimates.” 
  A final note about the appropriateness of this topic for the journal’s scope.  We think most remote sensing researchers consider validation an integral component of the scientific process, and we assert that a paper discussing practical validation issues is of inherent interest to the field.  Further, the description of this special issues includes this sentence: We encourage applications tackling issues of integrating ground and satellite data for calibration and validation of remote sensing-based biomass observations.  This is precisely the intention of this Technical Note.
The following comments taken from the reviewer's annotated copy of the manuscript  
(line 16) But what about the uncertainty of biomass estimates obtained by remote sensing?  To that end, it seems to me that only aspects of the reference uncertainty can be assessed!  As now clarified throughout the Introduction, including the text cited above, this paper introduces a dataset designed to validate (understand the uncertainty of) remotely sensed estimates.  Our analysis of the uncertainty of the reference data should lead to its more informed and more appropriate use of the validation data we document.
(Line 72)  NT is expected to report ... "new developments, significant advances and novel aspects of experimental and theoretical methods and techniques RELEVANT to the scope of Remote Sensing.  "Put that way, the relevance of the contribution is not characterized by the word SIMPLY ...  On the other hand, isn't the second purpose too abstract? Without an experiment that validates the spatial database, I mean. We removed the word "simply."  Here and elsewhere, the reviewer mistakenly states that introduction of a remotely sensed model/map would "validate" our reference dataset.  Reference data can be used to validate modeled predictions; the reverse (as suggested here and several places below) is not possible.
(line 106)  This is very relevant, but it needs to be validated. The discussion cannot be supported only by the great potential of the benchmark. See above about the reviewer's use of the term validation.  Regarding the paper's scope, we hope that revisions throughout the Introduction now make it clearer that we used our reference dataset to highlight potential discrepancies between mapped and reference data (which we accommodate in our reference dataset through inclusion of estimates using several combinations of carbon pools and allometric approaches).
(Line 120) Very confused. A flowchart of the activities carried out could better clarify the procedure adopted.  Each EMAP hex has 24.3 km2? Did the tessellation produce 12591 hexagons? Although this information is in the introduction, it could be transferred here, as it would make it easier to understand the method. We can see how the reader would be confused by our prior description.  We have now omitted reference to the second level of hexes, which was unnecessary.  We now emphasize only that there are approximately 27 field plots in each of the hexagons we used as areas of interest.
(line 132) As this aspect is important to verify the robustness of the reference elaboration process, it needs to be presented more clearly. The methodology needs to be better systematized. This paragraph (lines 153-160) is very important for this understanding and must be detailed. This and the following 3 comments request more information about FIA's methods and statistical estimators.  We have added an additional paragraph in this region of the paper, and we have enhanced methodological detail that was already present in the cited paragraph.  We also point out that the fifth and sixth paragraphs of the Introduction as well as the entirety of Section 2.2 focus on the design and use of FIA (the US national forest inventory).  It is certainly true that we have not fully enumerated all details of FIA data collection and analysis, but we feel that the level of detail now provided is appropriate for a Technical Note, especially in light of several citations to source material. 
(line 136) Was this procedure all developed by the authors in order to build the reference base? Clarified that these are standard FIA variables.  Reference to FIA source material now provided.
(line 153)  If the primary objective of TN is to report the construction of the spatial database of the biomass estimate then this process, with all the variables involved, must be described in detail. Not in a single paragraph, with several attributes that are unknown, equations, etc.  What variables and equations are used to estimate each attribute? this information must be included in the methodology and can be presented in Table 1 See above response.
(line 155) A map of this variable is presented in the results. It is necessary to clarify how it is estimated (from what measures?). These statements were added in a much-revised section:
Allometry under these three variations is specific by species or species group.  The first set of models, based on tree diameter measurements alone, was published by Jenkins et al. (2004) and is here called the “Jenkins” allometry. 
-and-
Applying different allometries to tree-level measurements creates varying plot-level measurements and propagates to statistical estimates of biomass at larger scales.  Here, we map differences in hex-level biomass estimates that are traceable to the difference between use of CRM and Jenkins allometries.
(line 167) EMAP hex? A line in the Introduction states that use of the term "hex" refers to EMAP hexes.  The reviewer is correct: the now-omitted reference to another level of hexagon made this term ambiguous.  Revisions summarized above eliminate this ambiguity.
(Table 1) Why not show the equation here or in the text? While the allometry of Jenkins et al. is fairly simple (just 10 equations for the entire country), detailed description of the other sets of allometric equations, which vary by species and sometimes by state, would require far more space than Remote Sensing allows.
(line 174) But how was the CRM calculated? What are the measures that compose it? See this earlier description in Section 2.2: A second approach, called the Component Ratio Method (CRM: (Woodall et al 2011)), is based upon FIA estimates of sound bole volume, with other tree components estimated as ratios (Jenkins et al 2004) of computed bole biomass.  CRM is currently used by FIA for national carbon reporting.  Unlike the Jenkins equations, CRM makes use of height measurements (through computation of volume).
(Line 226) But this is not here! It would not be the case to validate this reference by selecting some areas, comparing the estimates with remotely sensed biomass maps. the discussion cannot be limited to that. The benchmark has not been validated. The reviewer's confusion about validating a reference dataset with a model has been addressed above.  However, we can see why the reviewer thinks the effect of allometry has not been addressed.  The previous legend of Figure 5 showed "CRM-Jenkins Biomass."  We intended this to mean the estimate from one allometry (CRM) minus the estimate from another (Jenkins), which is what the revised legend specifies.  We acknowledge that this ambiguity may have led to the reviewer to miss this element of the results.  Our previous methodological description was also insufficient.  Among the clarifications, we added: "Applying different allometries to tree-level measurements creates varying plot-level measurements, and that variance propagates to statistical estimates of biomass at larger scales.  Here, we map differences in hex-level biomass estimates that are traceable to the difference between use of CRM and Jenkins allometries."
(line 269) Shouldn't that information be included in the introduction? Quite right.  We have moved this to the Introduction.

Reviewer 3 Report

This research aim is the understanding of the uncertainty of remotely sensed forest biomass estimation with a field-based benchmark product for the united states. the paper reported production of a map of mean biomass estimates created at approximately the finest scale (64000 ha) allowed by FIA sample density. Here are my comments for reviewing this manuscript. 

However, the manuscript has a lack of methodologies to present how remotely sensed data were analyzed. Also, the contents of the manuscript are not matched with the Remote sense journal article. 

Also, it is hard to find the validation process of the result of the allometric equation. If you want to validate your result of biomass estimation, you need to compare your research result with field study or any other estimation methods to prove your research accuracy. 

Based on the above issues, this article needs major revision and submission to a forest-related journal. 

Author Response

Comment Revision
This research aim is the understanding of the uncertainty of remotely sensed forest biomass estimation with a field-based benchmark product for the united states. the paper reported production of a map of mean biomass estimates created at approximately the finest scale (64000 ha) allowed by FIA sample density. Here are my comments for reviewing this manuscript. However, the manuscript has a lack of methodologies to present how remotely sensed data were analyzed.  Remotely sensed data were not analyzed.  Forest inventory data were analyzed in a way that highlights critical needs for the validation of remotely sensed products.  We acknowledge that the goals and message of the paper were not sufficiently clear in the original submission.  Both 1) introduction of a reference dataset and 2) analysis of the uncertainty involved with validating remotely sensed product using ground data are goals of this paper.  Several alterations to the Introduction make this more clear.  For example, we added this sentence: This Technical Note documents an alternative use of inventory data for validation.  We produced a map of local-scale statistical estimates, based solely on a designed sample of ground measurements, that may be of use in evaluating the myriad remotely sensed biomass maps that will soon be in circulation.
We also edited this statement: "These “benchmark” estimates are considered more authoritative than the remotely sensed estimates to which they may be compared because they are based solely upon straightforward sample theory and quality-controlled field measurements instead of models using auxiliary data.  This is not to say, however, that the inventory-based benchmark is without uncertainty. The second goal of this paper, beyond documenting an FIA-based reference dataset, is to illuminate concurrent sources of uncertainty that complicate the task of assessing the error of a remotely sensed biomass map."  Further, we re-wrote much of both the Abstract and the Conclusion sections to better articulate the what we learned about potential inventory-map discrepancies.  Lastly, we changed the title to: “A comprehensive forest biomass dataset for the US allows customized validation of remotely sensed biomass estimates.” 
Also, the contents of the manuscript are not matched with the Remote sense journal article.  We think most remote sensing researchers consider validation an integral component of the scientific process, and we assert that a paper discussing practical validation issues is of inherent interest to the field of remote sensing.  Further, the description of this special issues includes this sentence: We encourage applications tackling issues of integrating ground and satellite data for calibration and validation of remote sensing-based biomass observations.  This is precisely the intention of this Technical Note.
Also, it is hard to find the validation process of the result of the allometric equation. If you want to validate your result of biomass estimation, you need to compare your research result with field study or any other estimation methods to prove your research accuracy.  We can see why the reviewer thinks the effect of allometry has not been addressed.  The previous legend of Figure 5 showed "CRM-Jenkins Biomass."  We intended this to mean the estimate from one allometry (CRM) minus the estimate from another (Jenkins), which is what the revised legend specifies.  We acknowledge that this ambiguity may have led the reviewer to miss this element of the results.  Our previous methodological description was also insufficient.  Among the clarifications, we added: "Applying different allometries to tree-level measurements creates varying plot-level measurements, and that variance propagates to statistical estimates of biomass at larger scales.  Here, we map differences in hex-level biomass estimates that are traceable to the difference between use of CRM and Jenkins allometries."

These statements were added in a much-revised section:
Allometry under these three variations is specific by species or species group.  The first set of models, based on tree diameter measurements alone, was published by Jenkins et al. (2004) and is here called the “Jenkins” allometry. 
-and-
Applying different allometries to tree-level measurements creates varying plot-level measurements and propagates to statistical estimates of biomass at larger scales.  Here, we map differences in hex-level biomass estimates that are traceable to the difference between use of CRM and Jenkins allometries.
Based on the above issues, this article needs major revision and submission to a forest-related journal.  We again assert that validation is an irreplaceable part of remote sensing science and that this paper will contribute to the appropriate validation of many remotely sensed biomass maps and tools.

Round 2

Reviewer 2 Report

The new title of the manuscript “A comprehensive forest biomass dataset for the US allows customized validation of remotely sensed biomass estimates” is more suited to its content. However, this content does not yet include the validation of remotely sensed biomass process based on the reference produced and documented in the manuscript.

In the previous review, it was suggested the development of an experiment that results in the biomass estimation by remote sensing from a test area and use of the database produced to validate the result of the experiment and make the manuscript more adherent to the scope of the journal. However, this was not considered, although the suggestions for changes in the text have been implemented.

I recognize the mistake made when referring to the need to validate the biomass estimates obtained by remote sensing from the database created as “database validation”, but I highlight the importance of including the validation of an estimate obtained by remote sensing. This would appropriately characterize the integration between ground and satellite data and the validation of remote sensing-based biomass estimates.

Author Response

We very much appreciated the reviewer’s perspective about the need to clarify what this paper accomplishes in addition to documenting a reference data set.  We had not been clear enough about our use of the reference dataset to highlight the need for careful map-reference alignment in specific dimensions (including: allometry, selection of carbon pools, forest/non-forest discrepancy).  The second half of the Abstract was re-written and reflects our efforts to improve in this area.  It is gratifying that the Reviewer recognizes that the requested changes in the text were implemented.

As the Reviewer notes, we did not take their suggestion to create a biomass map for the purpose of illustrating the comparison of modeled and reference biomass estimates.  In our view, this is beyond the scope of our Technical Note, and documenting the mapping process (in addition to evaluating the accuracy of a specific product) would divert the reader’s attention from our primary topic: the process of validation.  We hope the dataset we document and the issues we raise will benefit international efforts to validate global remotely sensed biomass products.  We further hope that the editors agree that this aspiration is appropriate for the special issue to which it was submitted.